# Lightweight Object Detection Algorithm for UAV Aerial Imagery

**DOI:** 10.3390/s23135786

**Published:** 2023-06-21

**Authors:** Jian Wang, Fei Zhang, Yuesong Zhang, Yahui Liu, Ting Cheng

**Affiliations:** 1Faculty of Information Engineering and Automation, Kunming University of Science and Technology, Kunming 650504, China; 20222104055@stu.kust.edu.cn (F.Z.); 20212204295@stu.kust.edu.cn (Y.Z.); 20222104064@stu.kust.edu.cn (Y.L.); 20222204326@stu.kust.edu.cn (T.C.); 2Yunnan Key Lab of Artificial Intelligence, Kunming University of Science and Technology, Kunming 650504, China

**Keywords:** UAV imagery, object detection, YOLOv5s, spatial pyramid pooling, loss function

## Abstract

Addressing the challenges of low detection precision and excessive parameter volume presented by the high resolution, significant scale variations, and complex backgrounds in UAV aerial imagery, this paper introduces MFP-YOLO, a lightweight detection algorithm based on YOLOv5s. Initially, a multipath inverse residual module is designed, and an attention mechanism is incorporated to manage the issues associated with significant scale variations and abundant interference from complex backgrounds. Then, parallel deconvolutional spatial pyramid pooling is employed to extract scale-specific information, enhancing multi-scale target detection. Furthermore, the Focal-*EIoU* loss function is utilized to augment the model’s focus on high-quality samples, consequently improving training stability and detection accuracy. Finally, a lightweight decoupled head replaces the original model’s detection head, accelerating network convergence speed and enhancing detection precision. Experimental results demonstrate that MFP-YOLO improved the mAP50 on the VisDrone 2019 validation and test sets by 12.9% and 8.0%, respectively, compared to the original YOLOv5s. At the same time, the model’s parameter volume and weight size were reduced by 79.2% and 73.7%, respectively, indicating that MFP-YOLO outperforms other mainstream algorithms in UAV aerial imagery detection tasks.

## 1. Introduction

With the ongoing advancement of aerial photography technology, unmanned aerial vehicles (UAVs) applied in a variety of mission scenarios, such as military reconnaissance [1], environmental monitoring [2], and traffic planning [3], are increasingly common. Factors such as the shooting angle, distance, and lighting conditions often lead to UAV aerial imagery with small target scales, large-scale variations, and dense distributions, thereby increasing the complexity of the background. Hence, accurate object detection in UAV aerial imagery poses significant challenges.

Traditional object detection algorithms, such as the sliding window-based method presented in [4] and the template matching-based method in [5], rely heavily on handcrafted features. This dependence limits their ability to handle background information and extract sufficient object features from complex scenarios. Furthermore, the need for multi-scale and multi-location detection increases computational complexity, proving to be a barrier to the development of lightweight object detection algorithms.

As deep learning technology continues to evolve, leveraging deep learning object detection algorithms for accurate and swift detection in drone aerial imagery has become a central research focus for many scholars. For instance, Reference [6] used a multi-level attention domain adaptation network for cross-regional oil palm tree counting and detection, ensuring high detection accuracy and reducing the necessity for manual annotation. Object detection algorithms based on deep learning can be classified broadly into two categories according to their detection processes. Firstly, two-stage detectors, represented by the R-CNN [7] series, initially generate candidate regions, which are then classified and localized. For instance, Reference [8] leveraged the Faster RCNN model in conjunction with the Refined Pyramid Feature (RPF) module and Hybrid Class Balance Loss module, evaluating their proposed model on three large-scale drone images to achieve satisfactory observation of individual oil palm growth statuses. However, due to the need for classifying and localizing numerous candidate regions, these detectors have relatively high computational complexity, making it challenging to meet real-time requirements. On the other hand, single-stage detectors such as the YOLO [9] series and SSD [10] eliminate the need for generating candidate regions instead directly executing classification and localization tasks on the feature map. For instance, Reference [11] proposed a lightweight aerial object detection algorithm (LAI-YOLOv5s) after improving three aspects: feature fusion path, detection head, backbone network, and model lightweighting. This algorithm not only incorporates a lightweight network model but also enhances the detection accuracy of aerial images in drone scenarios. Since the generation and screening process of candidate regions is eliminated, although the accuracy is less than two-stage detectors, it shows significant advantages in computational efficiency. Particularly, single-stage detectors such as YOLOv5s, which maintain a balance between detection accuracy, speed, and model size, are better suited for object detection tasks in drone aerial scenes.

In aerial images captured by unmanned aerial vehicles (UAVs), the small size of targets, significant scale variations, and the prevalence of complex interference backgrounds pose unique challenges to object detection algorithms. To tackle these challenges, this paper presents MFP-YOLO, a lightweight detection algorithm based on YOLOv5s. This algorithm significantly reduces the model’s size while enhancing detection accuracy, thereby achieving a lightweight model and high-precision target detection.

The main contributions of this paper include:

(1) By designing a Multipath Inverted Residual Block (MIRB) and incorporating the CBAM attention mechanism, we have effectively addressed the problems of large-scale target variation and the difficulty of detecting small targets within complex backgrounds in UAV aerial images.

(2) By transforming the original Spatial Pyramid Pooling framework in YOLOv5s into the Parallel Involution Spatial Pyramid Pooling Framework (PISPPF), we can extract object information at specific scales, thereby enhancing the detection capability of multi-scale targets in UAV aerial images.

(3) The incorporation of the Focal-*EIoU* loss function has improved the model’s focus on high-quality samples in UAV aerial images, simultaneously enhancing model training stability and detection accuracy.

(4) We have replaced the original YOLOv5s’ coupled detection head with a lightweight decoupled one. This allows the decoupling of the tasks of classification, regression, and confidence scoring in object detection, thereby accelerating the model’s convergence speed and enhancing detection accuracy.

Experimental results show that MFP-YOLO improves the mAP50 on the VisDrone 2019 validation and test sets by 12.9% and 8.0%, respectively, compared to the original YOLOv5s. Simultaneously, the size of the model parameters and weights have been reduced by 79.2% and 73.7%, respectively. In comparison to other mainstream detection algorithms, MFP-YOLO demonstrates superior performance in UAV aerial image detection tasks.

The paper is organized into five sections: In Section 2, we introduce the YOLOv5 object detection algorithm. In Section 3, we elaborate on the proposed method. In Section 4, we present the experimental process and discuss the results, which include comparative and ablation experiments, as well as a comparative analysis of experimental results. Section 5 concludes the paper.

## 2. YOLOv5s Baseline Algorithm

Propelled by advancements in deep learning, the single-stage detector YOLOv5s exhibits substantial improvements over its predecessors. It achieves a delicate balance in detection accuracy, inference speed, and model size through data augmentation, adaptive prior boxes, and selecting different model sizes. Table 1 illustrates the performance metrics of various detectors on the COCO val 2017 dataset. As shown in Table 1, YOLOv5s excels in the number of parameters and detection frame rates, effectively balancing various metrics. Specifically, for drone-oriented scenarios, YOLOv5s stands out as the most appropriate real-time lightweight detection algorithm.

Compared to other detection algorithms, YOLOv5s has the following distinctive characteristics:

### 2.1. Feature Extraction

Feature extraction in YOLOv5s primarily relies on the C3 module and the SPPF structure. The convolutional and Bottleneck structures within the C3 module facilitate effective information propagation across different levels, reducing model complexity, ensuring model performance, and enhancing computational efficiency. The SPPF, consisting of convolution layers and maximum pooling operations, aims to effectively extract multi-scale features while retaining more contextual information.

### 2.2. Feature Fusion

The feature fusion in YOLOv5s predominantly leverages the Feature Pyramid Network (FPN) [12] and Path Aggregation Network (PAN) [13] for multi-scale feature fusion. The FPN disseminates high-level features to lower levels in a top-down manner and establishes lateral connections in a bottom-up approach. Building upon the FPN, the PAN facilitates a richer information flow by passing high-level features to lower layers in a top-down manner while up-sampling low-level features to higher levels in a bottom-up approach. This bi-directional information flow design enhances the network’s detection capabilities for multi-scale objects.

### 2.3. Output Detection

For output detection, YOLOv5s employs the multi-scale prediction strategy from YOLOv3 [14], decoding the feature maps, post-fusion, into the final detection results. At every scale, predictive bounding boxes are generated via the mechanism of anchor points and the technique of bounding box regression. Each preset anchor point is evaluated to check if it encompasses an object. If the presence of an object is confirmed, the object’s category is further predicted. In the end, non-maximum suppression is utilized to sift through the prediction results, selecting the targets with high confidence levels.

### 2.4. Loss Function

YOLOv5s addresses the challenges posed by cross-scale object detection and the ratio disparity between the dimensions of predicted and actual bounding boxes by utilizing the Complete Intersection over Union (*CIoU*) loss function. This function takes into account factors such as distance, aspect ratio, overlap rate, and scale parameters, thereby ensuring a more stable regression of bounding boxes.

The formula for calculating the *CIoU* loss function is presented in Equation (1).
(1)$$CIoU=1-IoU+\frac{\rho^{2}\left(p,p^{gt}\right)}{c^{2}}+\alpha V$$

The calculation formulas for *α* (a tunable hyperparameter) and *V* (a parameter that gauges the similarity in aspect ratio between the predicted and the actual boxes) are shown in Equations (2) and (3), respectively.
(2)$$\alpha =\{\begin{array}{ccc} 0 & , & IoU<0.5\\ \left(1-IoU\right)+V & , & IoU\geq 0.5 \end{array}$$
(3)$$V=\frac{4}{\pi^{2}}{\left(\arctan\frac{w^{gt}}{h^{gt}}-\arctan\frac{w}{h}\right)}^{2}$$

## 3. Improved YOLOv5s Model: MFP-YOLO

YOLOv5s exhibits robust performance in diverse detection settings. However, its effectiveness in object detection within drone-based aerial imagery is suboptimal. Compared to high-accuracy detection models such as YOLOv3 and lightweight counterparts such as YOLOv7-Tiny [15], YOLOv5s presents notable deficiencies in detection precision and model size. To tackle these shortcomings, this study concentrates on refining four critical components of YOLOv5s: feature extraction, feature fusion, detection head, and loss function. In response to the specific detection demands of drone aerial imagery, we propose MFP-YOLO, an optimized lightweight, high-precision real-time detection algorithm.

### 3.1. MIRB + C3_CBAM Module

Despite the C3 module enhancing feature extraction ability by using the Bottleneck structure, it still struggles with the challenges of large target scale variations and complex backgrounds present in drone aerial imagery. Furthermore, the C3 module treats channel and positional information uniformly across all feature maps, leading to an unnecessary introduction of significant noise. To combat these issues, our research introduces two primary improvement strategies:

Firstly, to address problems of extensive target scale variation and difficulties in feature extraction and fusion, we have devised the Multipath Inverted Residual Block (MIRB), as illustrated in Figure 1. This module adopts multi-channel parallel processing operations, dividing the feature map and separately applying the Inverted Residual structure [16], as represented in Figure 2. The architecture expands the number of input channels, employs depth-wise separable convolutions, and finally reduces the channel count to enhance feature expression while maintaining a low computational cost. The MIRB module processes four segments of the feature map in parallel—one segment is passed directly to conserve the original information, while the remaining three are processed using the inverted residual structure and summed with the previous branch’s output. This configuration enables the fusion of features from different layers and the learning of high-level abstract features, satisfying the requirement for multi-scale variation information in drone imagery. Parallel processing operations not only lessen computational burden but also improve feature processing efficiency and prevent gradient vanishing, making the approach suitable for resource-limited devices such as drones, thus fulfilling the dual requirement of real-time processing and accuracy.

Secondly, considering the challenges faced by the C3 module in dealing with dense small targets in complex backgrounds, this paper incorporates the Convolutional Block Attention Module (CBAM) attention mechanism [17] into the existing C3 module, creating the C3_CBAM module as shown in Figure 2. Moreover, the structure diagram of the CBAM module is shown in Figure 3. The CBAM enhances the network’s response to crucial features by implementing spatial and channel attention mechanisms. The inclusion of CBAM allows the C3_CBAM module to strengthen the detection of small targets using channel attention, while spatial attention enables the model to focus on significant areas within complex environments, thus improving detection accuracy. Additionally, the module accomplishes efficient parallel computation by leveraging global pooling and convolution operations, meeting the demands for real-time drone target detection.

In summary, to overcome the limitations of the C3 module, this study introduces two improved modules: MIRB and C3_CBAM. These modules significantly enhance the model’s capacity to detect multi-scale targets and dense small targets in complex backgrounds within drone aerial imagery while maintaining computational efficiency.

### 3.2. PISPPF Feature Pyramid Module

The SPPF method conducts multi-scale feature extraction via consecutive max pooling and feature concatenation. However, two issues arise from this strategy. First, recurrent serial pooling can lead to the propagation of erroneous information and the loss of small target features. Second, the direct concatenation neglects the intricate inter-scale feature relationships, posing challenges for multi-scale targets and high-resolution images, thereby impacting the accuracy of the model’s detection.

To mitigate these limitations inherent to the original SPPF, this study introduces the Parallel Involution Spatial Pyramid Pooling Framework (PISPPF), as depicted in Figure 4. Initial feature extraction is undertaken through standard convolution, after which the outcome is transferred into multiple parallel modules for further branch processing. Each branch consists of a max pooling operation, an involution, as illustrated in Figure 5, and a standard convolution operation [18]. Subsequently, through feature concatenation and another standard convolution, the final feature map is derived. Notably, PISPPF, as compared to SPPF, innovates by employing parallel operations, modifying the activation function, integrating involution structures, and adding an extra standard convolution operation prior to feature concatenation. The specific procedural details are as follows:

Initially, the parallel branch operations help alleviate the feature loss and error information propagation caused by serial pooling while enhancing the model’s efficiency. Second, the activation function is replaced with LeakyReLU, ensuring training stability and diminishing the computational burden. Furthermore, an involution structure is implemented to extract pixel information at specific scales. As a form of dynamic convolution, involution adaptively adjusts the weights of convolutional kernels to align with the input characteristics, effectively extracting specific scale information and addressing the challenges associated with substantial variations in target scale and difficulties in small target feature extraction. Specifically, the involution converts the input feature map into a 1 × 1 × k^2^ feature map via a 1 × 1 convolution. This feature map is then reshaped into a K × K × 1 convolutional kernel and applied to each channel of the input feature map, yielding a K × K × C feature map. The resulting feature map is further subjected to a 1 × 1 convolution operation to produce a 1 × 1 × K feature map.

Lastly, an additional standard convolution operation is conducted prior to feature concatenation in each branch. This operation ensures that the concatenated feature map comprises solely the specific scale information extracted by the involution, thereby eliminating redundant feature information.

Of note is the fact that the computational load of involution is primarily determined by the size of the kernel rather than the input image size. This attribute is particularly beneficial for high-resolution images as it leads to a reduction in resource consumption and an enhancement in the model’s performance.

### 3.3. Adjustment and Optimization of Loss Functions

Despite the strong performance of the *CIoU* loss function in handling cross-scale target detection and the ratio of predicted to actual bounding boxes, it struggles when dealing with variations in bounding box sizes and issues of sample imbalance. Therefore, this study introduces the *EIoU* [19] and FocalL1 loss functions for further optimization and adjustment.

To address the limitation of the *CIoU* loss function that cannot distinguish bounding boxes with the same centers and aspect ratios but different sizes, we employ the *EIoU* loss function. This function modifies the original aspect ratio into a width-height value regression, transforming the original formula: *CIoU* = *IoU* + center point loss + aspect ratio loss; into *EIoU* = *IoU* + center point loss + width loss + height loss. The calculation for the *EIoU* loss function is outlined in Equation (4).
(4)$$CIoU=1-IoU+\frac{\rho^{2}\left(p,p^{gt}\right)}{c^{2}}+\frac{\rho^{2}\left(w,w^{gt}\right)}{c_{w}^{2}}+\frac{\rho^{2}\left(h,h^{gt}\right)}{c_{h}^{2}}$$

Furthermore, to tackle the problem of sample imbalance, we implement the FocalL1 loss function. This function, derived from Focal Loss, adjusts the weight parameters to regulate the attention given to samples of different quality. It assigns lesser weights to lower-quality samples and greater weights to higher-quality samples, enabling the model to focus more on the high-quality samples, thus enhancing the training process.

In the end, the *EIoU* and FocalL1 loss functions are combined to form Focal-*EIoU*. This composite loss function not only achieves a more precise match between predicted and ground truth boxes but also effectively addresses the issue of sample imbalance. It better meets the demands of unmanned aerial vehicle (UAV) image detection tasks. The computation for Focal-*EIoU* is presented in Equation (5).
(5)$$\mathrm{Focal}-EIoU=IoU^{\gamma }*EIoU$$

Herein, the parameter *γ* is utilized to adjust the focus of the loss function and to modulate the attention paid to samples of varied quality.

### 3.4. Decoupling of Hybrid Convolution Module

In the field of object detection, an inherent conflict exists among the tasks of classification, regression, and confidence estimation. As shown in Figure 6, the detection head of YOLOv5 is still coupled. However, research from YOLOX [20] has demonstrated that decoupling these tasks can significantly enhance the network’s convergence speed and detection accuracy.

Each prediction head in YOLOX contains several fully connected convolution layers, causing each layer’s channel count to accumulate independently and, consequently, a substantial increase in the total parameters. Considering the balance between detection performances, this paper designs a lightweight Partial Shuffle Decouple (PSD) head, as depicted in Figure 7.

Initially, the input features undergo 1 × 1 convolution to reduce the channel dimension, which alleviates the computational complexity of subsequent operations. Next, before feature branching, a shared Partial Shuffle module is added, as shown in Figure 8. This addition obviates the need for independently and repetitively processing feature information in later branches, thereby further decreasing the model parameters. Finally, the Partial Shuffle module is utilized within the classification, regression, and object confidence branches. It takes advantage of the Partial Convolution [21] and draws on the Channel Shuffle design [22], both of which greatly diminish computational complexity while maintaining accuracy.

Partial Convolution applies filters to a part of the input channels, reducing redundant convolution computations and memory access and, thus, boosting computational efficiency. However, this restricts information exchange among channels. The structure of Partial Convolution is illustrated in Figure 9. Conversely, Channel Shuffle enables channel information sharing between adjacent network layers by reorganizing the feature map along the channel dimension. By establishing the interconnectedness of these two modules, the computational efficiency and feature redundancy reduction of Partial Convolution is maintained, while the correlation between the upper and lower channels is strengthened, thereby improving the model’s comprehension and utilization of features.

In conclusion, the PSD, by reducing the number of channels and incorporating strategies such as Partial Convolution and Channel Shuffle, preserves the balance of model detection performance while reducing the overall number of parameters.

### 3.5. Overall Model Architecture

MFP-YOLO is primarily devised to confront the challenges presented by a broad range of targets, intricate backgrounds, and amplified model weights associated with an increased number of layers in unmanned aerial vehicle (UAV) imagery. This paper redesigns the network architecture and streamlines the network model.

Initially, all C3 modules in the original YOLOv5s are replaced with MIRB and C3_CBAM modules. The backbone network incorporates two pairs of “MIRB and C3_CBAM”, enhancing the extraction of deep features. Conversely, the neck network diminishes the layer count and complexity, concentrating on detecting small and densely distributed objects whilst minimizing the network model size. Subsequently, the Parallel Inverted Spatial Pyramid Pooling with Deconvolution Fusion (PISPPF) structure is introduced to bolster feature reuse extraction and guarantee stability during the training process. Further, the coupled detection head in the original model is decoupled, incorporating the lightweight Partial Shuffle Decouple (PCD) head. This approach not only balances model size but also improves detection accuracy. Finally, the Focal-*CIoU* loss function supplants the original *CIoU* loss function, addressing bounding box size disparities and alleviating the imbalance between high-quality and low-quality samples, consequently augmenting model convergence speed and detection accuracy. Figure 10 illustrates the architecture of the MFP-YOLO network.

## 4. Experimental Results and Analysis

### 4.1. Experimental Setup

#### 4.1.1. VisDrone2019 Dataset

The present study utilizes the VisDrone2019 dataset [23], released by the AISKYEYE team at Tianjin University. This dataset encompasses all-weather data sourced from diverse environments in 14 Chinese cities captured by drones at various altitudes. After cropping and processing, the dataset produced 10,209 static images with resolutions ranging from 960 × 540 to 2000 × 1500, thus providing a comprehensive and extensive data source. These images are divided into 6471 training samples, 548 validation samples, and 3190 testing samples, collectively containing approximately 2.6 million target instances.

The dataset includes a wide variety of object categories, encompassing pedestrians (either walking or standing), individuals in other postures, cars, vans, buses, trucks, motorcycles, bicycles, awning tricycles, and other types of tricycles. The diversity in object size, category, and image resolution significantly contributes to the complexity and richness of the dataset, providing an ideal data foundation for this study on drone aerial imagery.

For the object detection tasks, objects whose aspect ratio is less than 0.1 of the original image size are categorized as small objects. As evident from Figure 11, the object size and class sample distribution in the VisDrone2019 dataset indicate that the majority of objects are within an aspect ratio of 0.1, with the class sample distribution being uneven. This characteristic aligns perfectly with the established object detection difficulty parameters set for drone aerial imaging in this study.

#### 4.1.2. Experimental Parameter Configuration

The experimental environment of this research incorporates the Windows 11 operating system, NVIDIA GeForce RTX 3060 GPU (manufactured by NVIDIA Corporation, Santa Clara, CA, USA) with 12 GB of VRAM,, and CUDA 11.7. The underlying experimental framework is Pytorch 1.8.1-based and utilizes the YOLOv5s, version 6.0. The experimental parameters include a batch size of 8, a learning rate of 0.01, an input size of 640, and training for 300 epochs. The training commences from scratch without leveraging any pre-trained weights, while other parameters remain default.

### 4.2. Performance Metrics

This study adopts the mean Average Precision (mAP) to evaluate the performance of the algorithmic model, where a higher mAP indicates superior model detection precision. The AP value is computed from the area enclosed by the Precision-Recall (PR) curve, formed during the training process, as depicted in Equation (6).
(6)$$AP_{IoU}=\int_{0}^{1}P(R)dR$$

*IoU*, the Intersection over Union between the detected and ground truth bounding boxes, is used alongside Precision (*P*) and Recall (*R*). These values are calculated from True Positive (*TP*), False Positive (*FP*), and False Negative (*FN*), as outlined in Equations (7) and (8), respectively.
(7)$$P=\frac{TP}{TP+FN}$$(8)$$R=\frac{TP}{TP+FP}$$

Upon completion of the training process, the model’s weight size (M), parameter count (M), and the number of floating-point operations (in GFLOPs, G) are output as the evaluative metrics for model complexity.

### 4.3. Experimental Results and Analysis

#### 4.3.1. Comparative Study on Loss Functions

This study aims to validate the efficacy of the improved loss function on the VisDrone dataset. We compared the detection accuracy subsequent to the modification of the loss function and devised experiments to identify the optimal *γ* parameter. *γ*, serving as a crucial hyperparameter, regulates the attention of the model toward high-quality samples. Optimizing *γ* enhances the predictive performance of the model. Pertinent experimental results are illustrated in Table 2.

When the *γ* hyperparameter of the loss function is set to 1.0, the model prioritizes high-quality samples. As depicted in Table 2, the model demonstrates the best performance when *γ* is set to 0.9, with an increase of 1.1% in mAP50. Consequently, this study establishes *γ* as 0.9 to achieve optimal detection results.

#### 4.3.2. Comparative Study on Decoupled Heads

The purpose of this section is to validate the equilibrium between various detection performance metrics attained by the proposed streamlined decoupled head, PCD. We conducted a comparative analysis between the YOLOv5s model with a coupled detection head, the YOLOv5s model augmented with the YOLOX decoupled head (Dhead), and the YOLOv5s model fitted with the proposed PCD decoupled head. Experimental outcomes are presented in Table 3.

As inferred from Table 3, the proposed PCD decoupled head incurs a marginal loss of 1.7% in accuracy compared to YOLOX’s decoupled head. However, the model size is reduced by 13.7 M, the number of floating-point operations decreases by 38.3 G, and there is a 0.5 ms reduction in inference time. Thus, while preserving acceptable accuracy levels, it successfully reduces model complexity and computational time, achieving an optimal balance between the efficiency and precision of the decoupled head.

#### 4.3.3. Comprehensive Ablation Study for Model Improvement

To rigorously assess the effects of the proposed improvements, a series of ablation experiments were performed by incrementally introducing enhancements into the baseline YOLOv5s model. These steps were designed to elucidate the specific contributions of each modification toward detection accuracy, model complexity, and inference time. The procedure was as follows: Initially, the C3 module was replaced by the MIRB + C3_CBAM module, reducing the overall number of modules; subsequently, the SPPF architecture was optimized to the PISPPF configuration; then, the original *CIoU* loss function was substituted by the Focal-*EIoU* loss function; finally, the conventional coupled detection head was replaced by the lightweight PCD decoupled head. The effects of each of these modifications were evaluated on the VisDrone2019 validation set. Detailed experimental results are provided in Table 4.

Table 4 demonstrates the impact of each modification on model performance. Beginning with the YOLOv5s baseline model, the substitution of the C3 module with MIRB + C3_CBAM resulted in a 7.6% increase in mAP50, a reduction in model parameter size to 1.03, and a decrease in weights to 2.76 M. This indicates that the MIRB + C3_CBAM configuration, with its parallel processing capabilities and introduction of attention mechanisms, effectively consolidates features from different layers and emphasizes key regions.

Figure 12 provides a comparison between the original YOLOv5s network model and the detection performance of the model when the C3 module is replaced with the MIRB + C3_CBAM module. The enhancement in the effective detection of complex targets in multiple backgrounds is due to the attention mechanism’s focus on target information in specific areas and the MIRB module’s integration of different scale target extractions.

Subsequently, the introduction of the PISPPF structure led to an increase in mAP50 to 43.8%, a slight increase in model parameters to 1.20, and weights to 3.3 M. Although this optimization led to a minor increase in parameters, it also led to a significant improvement in detection accuracy, validating the effectiveness of convolutions in extracting information at specific scales. Then, the application of the Focal-*EIoU* loss function resulted in a mere 0.7% increase in mAP50, but without any added complexity to the model. This highlights the efficacy of the Focal-*EIoU* loss function in addressing class imbalance and refining bounding box sizes. 

Finally, we introduced the PCD decoupling head, which increased the model’s mAP50 to 46.0%. Although the complexity of the model has increased, the introduction of the PCD decoupling head achieved task decoupling, which enhanced detection accuracy while preserving the model’s complexity, resulting in an accuracy improvement of 1.5%. More importantly, the PCD decoupling head significantly expedited the network’s convergence speed and improved its detection accuracy. Figure 13 presents a comparison of detection accuracy curves before and after the addition of the PCD decoupling head. During the training process, the model with the decoupling head (indicated by the red line) converged faster than the model without the decoupling head (indicated by the green line). Thus, the inclusion of the PCD decoupling head not only hastened the network’s convergence but also reduced the training time, thereby enhancing the overall detection accuracy of the model.

In conclusion, the proposed improvements resulted in a 12.9% increase in mAP50 while simultaneously reducing the parameter size and model size to 1.48 and 3.81, respectively. This effectively improved detection performance while reducing the model footprint, thereby demonstrating the practicality and efficacy of the proposed improvements.

#### 4.3.4. Performance Benchmarking of Different Algorithms on VisDrone 2019 Validation and Test Sets

The VisDrone 2019 dataset primarily contains high-resolution images, with resolutions ranging from 960 to 2000. Utilizing the original YOLOv5s model (with a default size of 640) compresses the images upon input, reducing the size of the targets and increasing the number of densely packed, small targets, thereby escalating the challenge of detection. To demonstrate the efficacy of our proposed algorithm, we conducted experiments at two different resolutions (a default size of 640 and an expanded size of 1024), comparing our refined model with other state-of-the-art detection algorithms. This comparison encompassed various performance metrics on both the validation and test sets, with comprehensive comparison results shown in Table 5.

Based on the data in Table 5, it is evident that as the input image size increases from 640 to 1024, there is an uptrend in the mAP50, the number of parameters, and the inference time across all algorithms. This outcome illustrates the influence of varying input image sizes on the detection performance. A larger input size affords more information to the algorithm, thereby enhancing object detection capabilities. However, it also increments the computational load per frame, resulting in slower processing speeds. Consequently, the optimal input size can be selected in practical applications, balancing the requirements for speed and precision.

For the default input size of 640, our MFP-YOLO algorithm is compared with notable high-accuracy detection algorithms such as YOLOv3, YOLOv5s, and YOLOv5x in terms of detection accuracy. MFP-YOLO outperforms these algorithms in mAP50 on both validation and test sets, registering an increase in accuracy of 6.8%, 12.9%, and 5.2% for the training sets and 3.2%, 8.0%, and 1.6% for the test sets, respectively. Despite the relatively marginal improvement over YOLOv5x, the weight size of MFP-YOLO is reduced by a staggering 97.8% in comparison to YOLOv5x. This corroborates that the MFP-YOLO algorithm significantly diminishes the model size while simultaneously enhancing detection accuracy.

In the context of model size, comparisons were drawn with other eminent compact detection algorithms such as YOLOv3-tiny, YOLOX-s, YOLOv7-tiny, and YOLOv8n. The MFP-YOLO algorithm’s parameter and weight sizes were considerably lower than these four counterparts, with reductions of 7.21 M, 7.52 M, 4.55 M, and 1.53 M; and 13.69 M, 5.13 M, 8.49 M, and 2.39 M in parameter size and weight size respectively. Even though the decrease in model size compared to YOLOv8n is relatively less pronounced, the validation and test sets’ mAP50 scores for MFP-YOLO are superior by 12.6% and 8.6%, respectively, compared to YOLOv8n. This observation again validates that the MFP-YOLO model effectively reduces the model size while improving detection accuracy.

In terms of inference speed, although no significant improvement is observed in comparison to other detection algorithms, the achieved average frame rate of 28.5 FPS still fulfills the real-time detection requirements of Unmanned Aerial Vehicle (UAV) scenarios. In essence, MFP-YOLO strikingly reduces the model size while sustaining high detection accuracy, and its inference speed is sufficient to accommodate the detection demands of practical scenarios, effectively performing the target detection task in UAV environments.

#### 4.3.5. Experimental Evaluation and Analysis

To further substantiate the effectiveness of the proposed algorithm, five sets of high-resolution images (with resolutions ranging from 1080 to 2000) from the VisDrone2019 dataset were selected. These images, characterized by complex backgrounds, dim lighting, small and densely-packed targets, and large target scales, were input into a model of size 640 for detection. The detection performance of various models is demonstrated in Figure 14, Figure 15, Figure 16, Figure 17 and Figure 18. The left side of each figure displays the results obtained by other algorithms, whereas the right side showcases the results yielded by the proposed MFP-YOLO algorithm. Despite these high-resolution images being resized to 640 for detection, which led to a further reduction in the target information, a comparative analysis of the detection results in Figure 14, Figure 15, Figure 16, Figure 17 and Figure 18 revealed:

(1) In Figure 14, the aerial image, taken from a high angle, features numerous small and densely-packed targets. The YOLOv5x model is able to detect the majority of the proximal targets but misses some pedestrians under the trees and vehicles arranged densely. Conversely, the MP-YOLOv5 network model exhibits superior performance, detecting pedestrians and vehicles under distant trees accurately without any missed or false detections.

(2) The aerial image in Figure 15 is captured from a completely vertical perspective, which not only reduces target features but also diminishes target sizes, posing challenges for accurate detection. While the YOLOv3 network model can accurately detect most large-scale targets, it misses a substantial number of small-scale targets, such as pedestrians and cyclists. In contrast, the MP-YOLOv5 network model maintains accuracy in large target detection and successfully identifies smaller or more complex targets, such as pedestrians and obstructed vehicles.

(3) The aerial image in Figure 16 is characterized by complex background features. Complications such as weather conditions and shooting angles render the background dim, increasing detection difficulty. The YOLOv7-tiny network model significantly misses targets and erroneously identifies non-existent classes, such as display boards, resulting in a significant amount of false and missed detections. However, the MP-YOLOv5 network model performs admirably under these conditions, accurately detecting target features in the dim background and successfully identifying the majority of targets without any false detections.

(4) The aerial image in Figure 17 poses a detection challenge due to the similarity of background features, high shooting angles, and long-distance photography. The YOLOv8n network model can accurately detect nearby targets while avoiding false detections but fails to identify a significant number of proximal small targets and vehicles on distant roads. Conversely, the MP-YOLOv5 network model demonstrates exemplary performance by accurately detecting all proximal targets with nearly no missed or false detections. It also accurately identifies vehicles on distant roads and assorted buses, resulting in the detection of more small and densely arranged targets and fewer missed or false detections.

(5) The aerial image in Figure 18 comprises numerous small and densely arranged targets amidst a complex background. The original YOLOv5s network model fails to identify a significant number of actual targets in the image. In contrast, the MP-YOLOv5 network model shows exceptional performance in this complex scenario, accurately detecting a multitude of vehicles driving under the distant overpass and successfully identifying pedestrians and vehicles densely distributed beneath a mall. As a result, it detects a higher number of targets and exhibits a lower rate of missed detections.

Based on the comparative graphs and analysis results of the above five different detection algorithms and the MFP-YOLO algorithm, it can be seen that the detection network proposed in this study exhibits superior detection performance in comparison to other models for aerial imagery, including high-resolution, densely packed small targets, multi-scale targets, dim scenarios, and complex backgrounds with multiple disturbances. This fully validates the effectiveness of the MFP-YOLO algorithm and highlights its practical value in conducting accurate and reliable target detection for drone aerial imagery. These findings also provide crucial references for further optimizing and enhancing the target detection techniques for drone aerial imagery.

## 5. Conclusions

In this study, in response to the unique properties of drone aerial imagery, we proposed the lightweight drone aerial image object detection algorithm MFP-YOLO. This algorithm takes into account strategies such as multi-scale feature fusion, feature extraction in complex backgrounds, sample imbalance, and decoupling of detection heads. It redesigns feature extraction and feature fusion, incorporates attention mechanisms to focus on important features, optimizes feature pyramid structures to extract specific target information, replaces loss functions for precise bounding box matching and to address sample imbalance issues, and decouples detection heads to balance model size and improve detection accuracy. The experiments with an input size of 640 on the VisDrone2019 dataset demonstrate that the mAP50 of the validation set and test set increased from 33.1% and 27.8% to 46.0% and 35.8%, respectively. The number of parameters and weight size decreased from 7.1 M and 14.5 M to 1.48 M and 3.81 M, respectively, meeting the lightweight requirements of the drone aerial image detection algorithm model. In comparison with other classic object detection algorithms, MFP-YOLO has significantly reduced the weight size and the number of model parameters while demonstrating high detection accuracy, satisfying the requirements for lightweight, high-accuracy, real-time detection in drone aerial imagery.

Furthermore, the findings of this study also suggest potential avenues for future work. Although there are certain similarities between aerial images captured by UAVs and remote sensing images, substantial differences exist in terms of imaging methods, image characteristics, and application objectives. Hence, further research and empirical validation are necessary to explore the transferability and adaptability of MFP-YOLO to other application scenarios, such as the classification of scenes in remote sensing images. In the future, we plan to explore methods such as TGRS-Partial Domain Adaptation for Scene Classification from Remote Sensing Imagery to uncover the potential application of MFP-YOLO in remote sensing image scene classification. This exploration will enhance our understanding of MFP-YOLO’s capabilities and open up new possibilities for the development and application of object detection technology in UAV aerial imagery.

In summary, the development of MFP-YOLO has introduced new perspectives and possibilities for object detection in UAV aerial imagery. Through further research, we anticipate expanding its applications and enabling its superior performance across a broader range of scenarios.

## Figures and Tables

**Figure 1 sensors-23-05786-f001:**
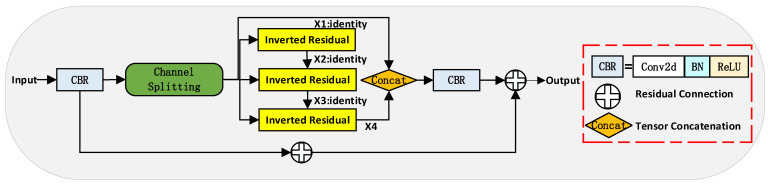
MIRB: Multipath inverse residual module.

**Figure 2 sensors-23-05786-f002:**
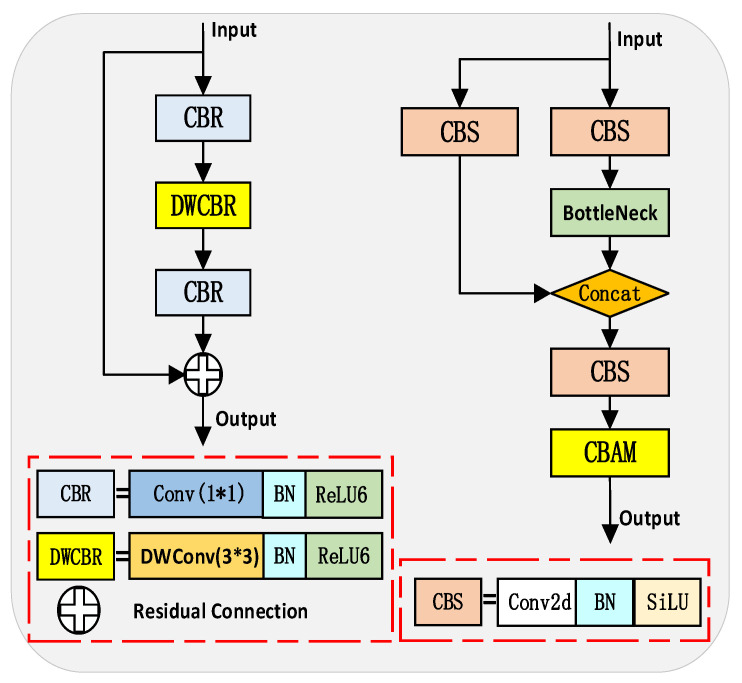
Inverted Residual: Inverted Residual Structure (**left**) and C3_CBAM Network Structure (**right**).

**Figure 3 sensors-23-05786-f003:**
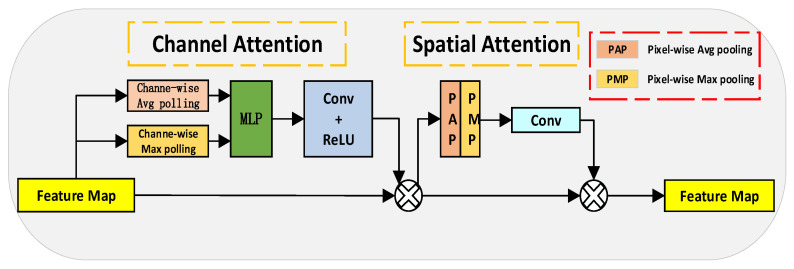
CBAM: Convolutional block attention module network structure.

**Figure 4 sensors-23-05786-f004:**
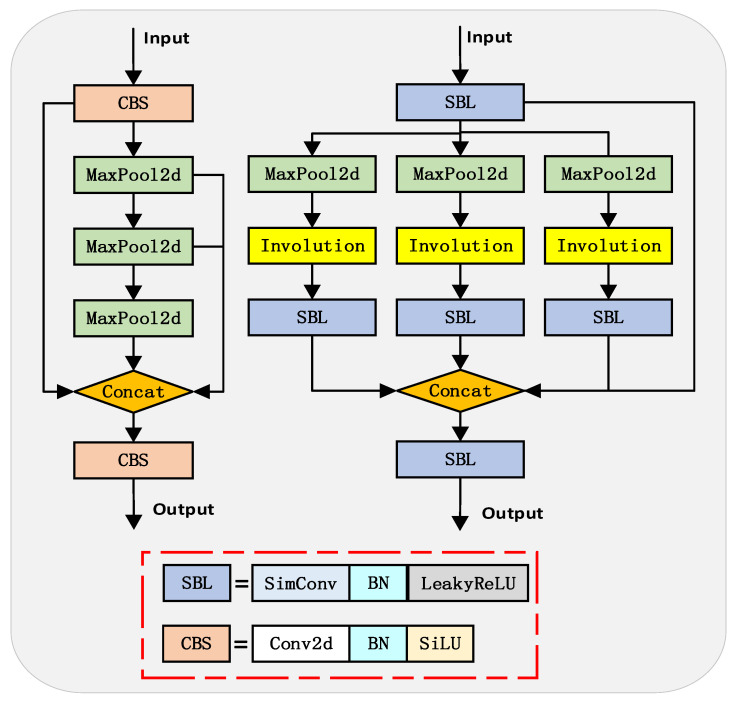
SPPF: Spatial pyramid pooling structure (**left**) and PISPPF: Parallel deconvolutional spatial pyramid pooling structure (**right**).

**Figure 5 sensors-23-05786-f005:**
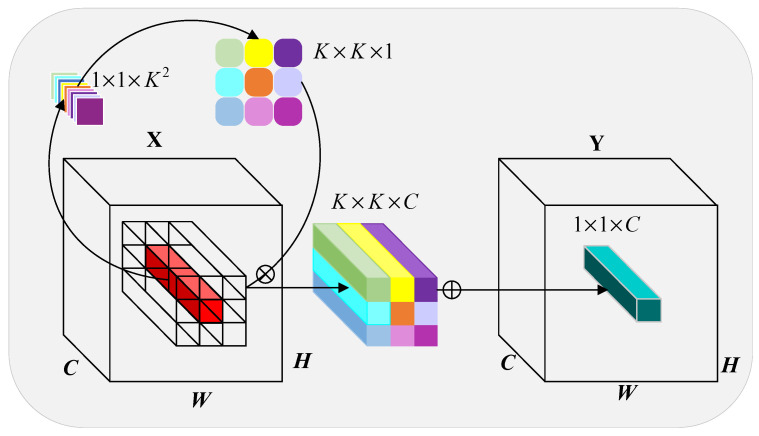
Involution: Involutional network model structure.

**Figure 6 sensors-23-05786-f006:**
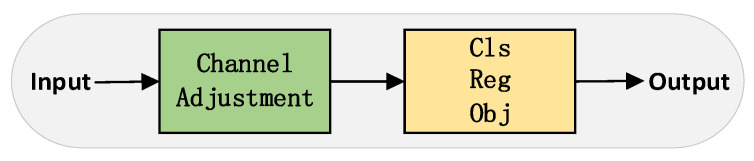
YOLOv5s: Coupled head structure diagram.

**Figure 7 sensors-23-05786-f007:**
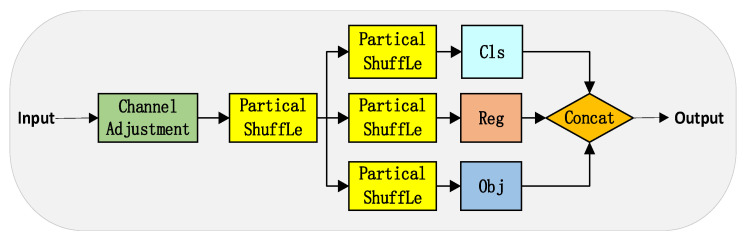
PSD: Partial shuffle decoupled head structure diagram.

**Figure 8 sensors-23-05786-f008:**
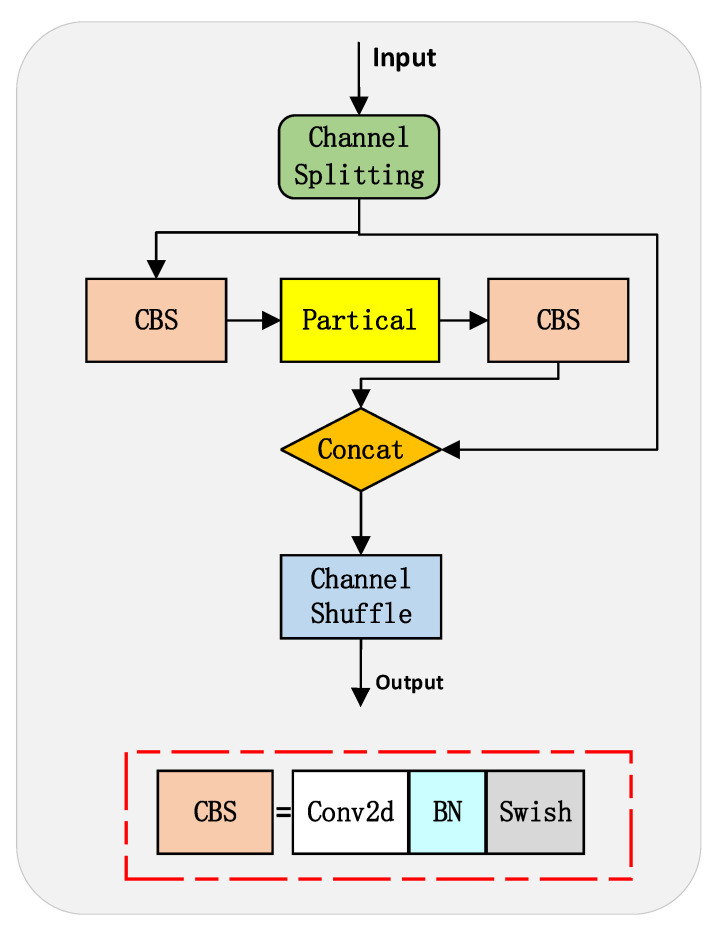
Partial shuffle network structure.

**Figure 9 sensors-23-05786-f009:**
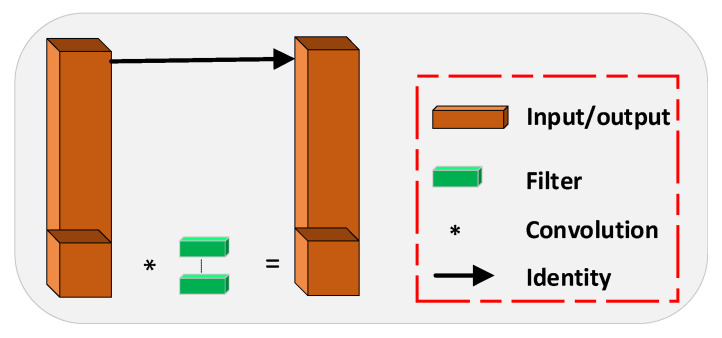
Partial convolutional network structure.

**Figure 10 sensors-23-05786-f010:**
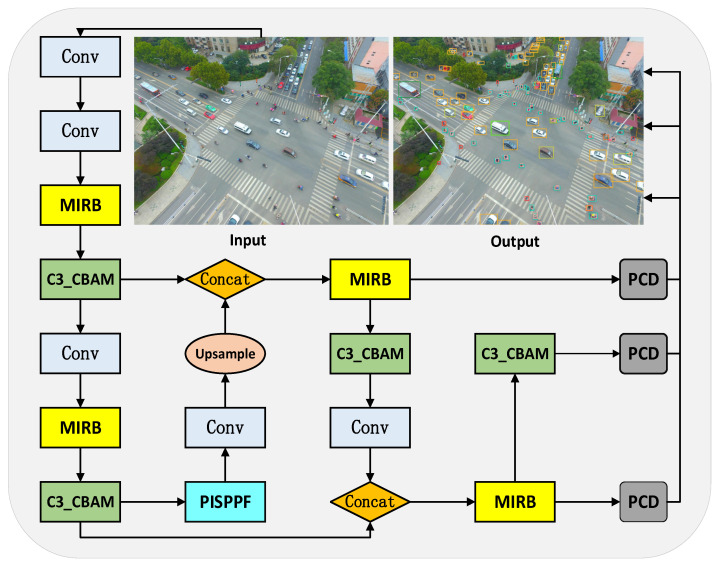
MFP-YOLO: Network model structure.

**Figure 11 sensors-23-05786-f011:**
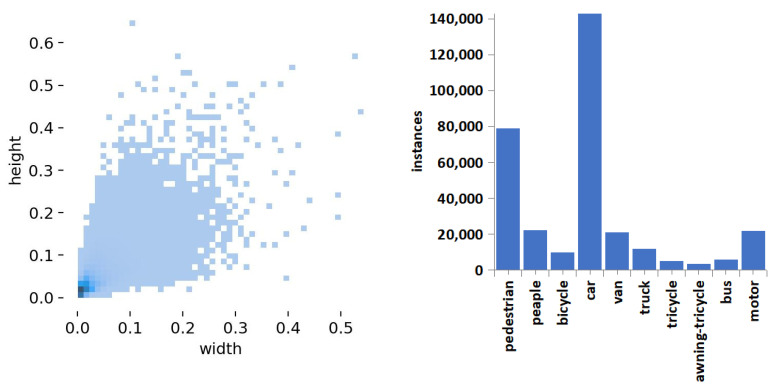
VisDrone2019 Dataset: Relative Size Distribution Statistics (**left**) and Class Sample Statistics (**right**).

**Figure 12 sensors-23-05786-f012:**
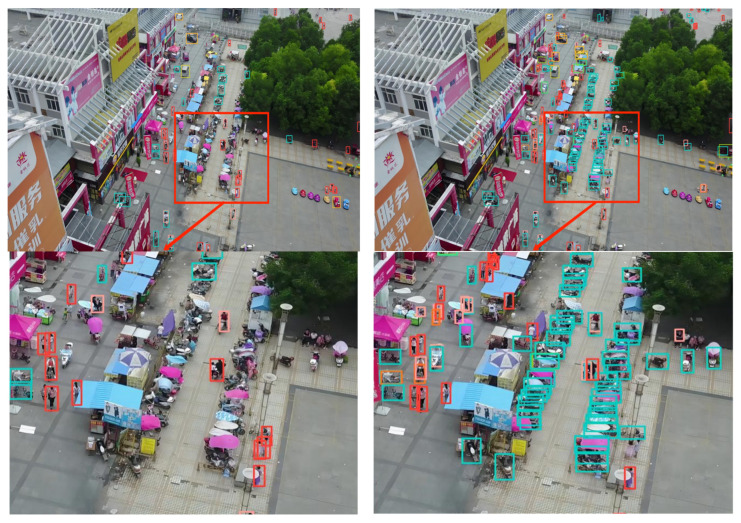
YOLOv5s vs. Algorithm After Replacing the C3 Module: Comparison of Detection Results (**left** and **right**, respectively).

**Figure 13 sensors-23-05786-f013:**
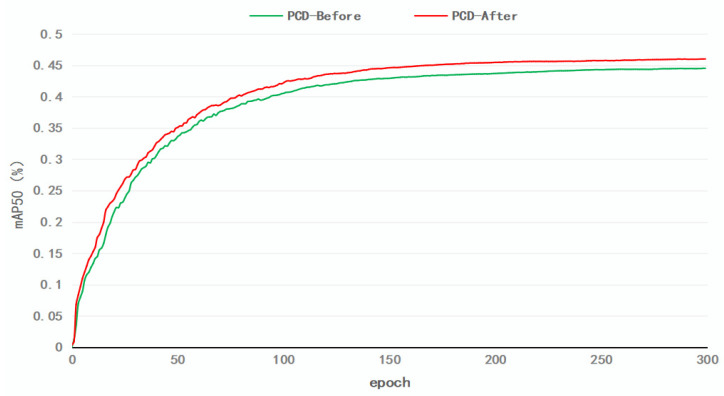
Comparison of Training mAP50 Curves Before and After the Addition of the PCD Module.

**Figure 14 sensors-23-05786-f014:**
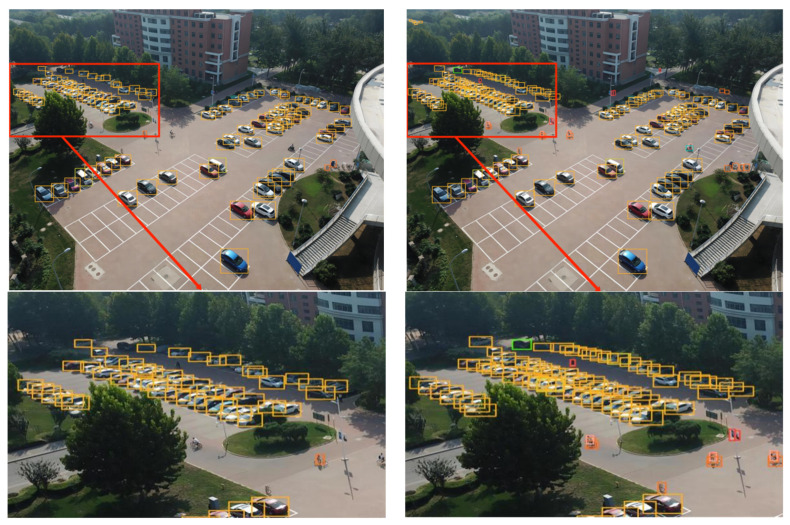
YOLOv5x vs. MFP-YOLO: Comparison of Detection Results (**left** and **right**, respectively).

**Figure 15 sensors-23-05786-f015:**
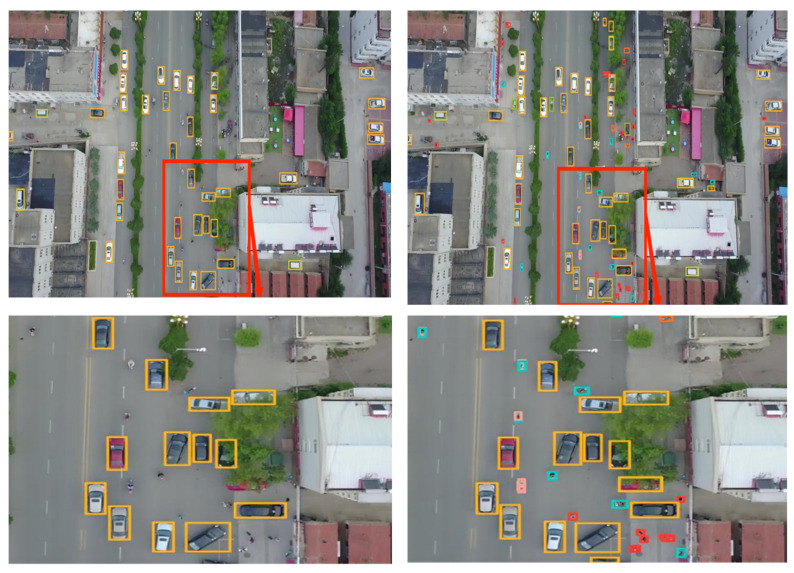
YOLOv3 vs. MFP-YOLO: Comparison of Detection Results (**left** and **right**, respectively).

**Figure 16 sensors-23-05786-f016:**
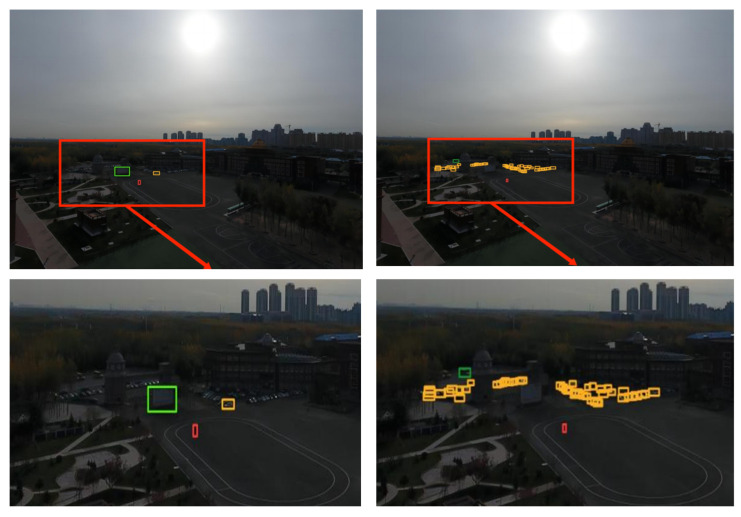
YOLOv7-tiny vs. MFP-YOLO: Comparison of Detection Results (**left** and **right**, respectively).

**Figure 17 sensors-23-05786-f017:**
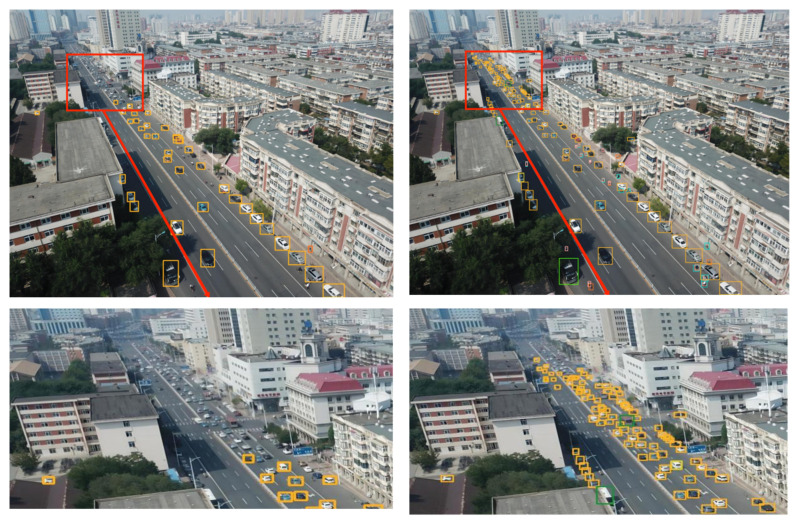
YOLOv8n vs. MFP-YOLO: Comparison of Detection Results (**left** and **right**, respectively).

**Figure 18 sensors-23-05786-f018:**
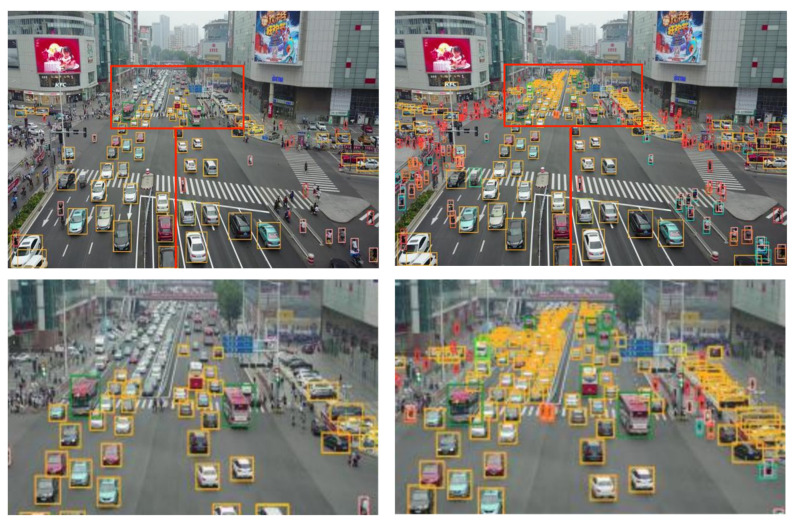
YOLOv5s vs. MFP-YOLO: Comparison of Detection Results (**left** and **right**, respectively).

**Table 1 sensors-23-05786-t001:** Performance comparison of several object detection algorithms.

Method	mAP50^val^ (%)	Params (M)	Speed (FPS)
Faster R-CNN	42.7	136.7	7
SSD300	41.2	23.7	46
YOLOv3	55.3	61.5	34
YOLOv5s	56.8	7.1	98

**Table 2 sensors-23-05786-t002:** Impact of changing loss function and parameters *γ* on mAP50.

Method	*γ*	mAP50^val^ (%)
YOLOv5s	-	33.1
Focal-*EIoU*	0.5	33.9
	0.7	34.0
	0.9	34.2
	1.0	33.8

**Table 3 sensors-23-05786-t003:** Performance comparison of detectors with different decoupling heads.

Method	mAP50^val^ (%)	GFLOPs (G)	Time (ms)	Size (M)
YOLOv5s	33.1	16.0	13.2	14.5
+Dhead	36.1	56.3	25	29.1
+PCD	34.4	18.0	24.5	15.4

**Table 4 sensors-23-05786-t004:** Ablation study comparison of different improvement methods.

Method	mAP50^val^ (%)	Params (M)	Size (M)
YOLOv5s	33.1	7.10	14.5
+C3 Replace	40.7	1.03	2.76
+PISPPF	43.8	1.20	3.3
+Loss function	44.5	1.20	3.3
+PCD	46.0	1.48	3.81

**Table 5 sensors-23-05786-t005:** Comparison of metrics for different detection algorithms on the VisDrone2019 dataset.

Method	Input Image Size	mAP50^val^ (%)	mAP50^test^ (%)	Params (M)	Size (M)	Time^val^ (ms)	Time^test^ (ms)	Average Speed/FPS
YOLOv3	640	39.2	32.6	61.55	123.6	25.2	19.2	22.2
	1024	51.9	43.1			38.5	32.2	35.35
YOLOv3-tiny	640	16.5	14.1	8.69	17.5	7.0	5.3	6.15
	1024	29.3	23.6			9.7	7.2	8.45
YOLOX-s	640	32.6	27.7	9.0	8.94	25.6	22.9	24.25
	1024	40.3	34.1			28.7	26.4	27.55
YOLOv7-tiny	640	35.4	30.3	6.03	12.3	24.8	23.6	24.2
	1024	45.2	38.1			26.8	25.4	26.1
YOLOv8n	640	33.4	27.2	3.01	6.2	12.9	6.2	9.55
	1024	43.4	35.9			14.1	7.4	10.75
YOLOv5s	640	33.1	27.8	7.10	14.5	13.2	11.8	12.5
	1024	45.1	38.0			17.8	16.0	16.9
YOLOv5x	640	40.8	34.2	86.28	173.2	28.1	27.0	27.55
	1024	53.2	43.9			64	62.9	63.45
MFP-YOLO	640	46.0	35.8	1.48	3.81	30.8	26.2	28.5
	1024	53.7	44.1			58	52.5	55.25

## Data Availability

The data presented in this study are available upon request from the corresponding author.

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
