# Peer review of "Lightweight Object Detection Algorithm for UAV Aerial Imagery"

_sensors, 2023, doi:10.3390/s23135786_

Round 1

Reviewer 1 Report

  1. Please highlight contributions and organization of this paper in the end of Sec. 1
  2. I recommend the authors add some discussion about the transferability of proposed MFP-YOLO, such as TGRS-Partial Domain Adaptation for Scene Classification from Remote Sensing Imagery
  3. I think the authors could add more visualization figures of different methods in Fig. 12. There are only two methods here. In addition, the authors could add some highlight regions using boxes or circles that MFP-YOLO performs better than other methods.
  4. I also think the authors could add some visualization figures for the results of Table 4.
  5. Some references are recommended to cite:
    • ISPRS-Growing status observation for oil palm trees using Unmanned Aerial Vehicle (UAV) images (object detection from UAV images)
    • SR-Lightweight aerial image object detection algorithm based on improved YOLOv5s (lightweight object detection using YOLO series)
    • ISPRS-Cross-regional oil palm tree counting and detection via a multi-level attention domain adaptation network (transfer learning for object detection using remote sensing images)

Reviewer 2 Report

Very interesting topic, well done and developed. Please I found problems reading the workfpows... due to the use of strong and dark colours. Please change the color of 1 to 10. TO read clearly wht is inside of some boxes I have to zoom too much. Use more "light" colours.

About Figure 11, it has been published in law resolution. Please extract a petter resolution image.

Reviewer 3 Report

i consider the topic addressed in the article well-described. I have no specific skills on the methods described, indeed i would recommend an extended review from the editorial board before considering for the final submission and acceptance.

Round 2

Reviewer 1 Report

The authors have addressed all my issues.